# Technology Acceptance Model for Exoskeletons for Rehabilitation of the Upper Limbs from Therapists’ Perspectives

**DOI:** 10.3390/s23031721

**Published:** 2023-02-03

**Authors:** Beatrice Luciani, Francesco Braghin, Alessandra Laura Giulia Pedrocchi, Marta Gandolla

**Affiliations:** 1Department of Mechanical Engineering, Politecnico di Milano, Via La Masa 1, 20156 Milano, Italy; 2NeuroEngineering And Medical Robotics Laboratory (NEARLab), Department of Electronics, Information and Bioengineering, Politecnico di Milano, Piazza Leonardo da Vinci, 32, 20133 Milano, Italy; 3WE-COBOT Lab, Politecnico di Milano, Polo Territoriale di Lecco, Via G. Previati, 1/c, 23900 Lecco, Italy

**Keywords:** technology acceptance model, rehabilitation exoskeletons, therapists, neuro-rehabilitation, multiple linear regression, Pearson’s correlation, integrated sensor systems

## Abstract

Over the last few years, exoskeletons have been demonstrated to be useful tools for supporting the execution of neuromotor rehabilitation sessions. However, they are still not very present in hospitals. Therapists tend to be wary of this type of technology, thus reducing its acceptability and, therefore, its everyday use in clinical practice. The work presented in this paper investigates a novel point of view that is different from that of patients, which is normally what is considered for similar analyses. Through the realization of a technology acceptance model, we investigate the factors that influence the acceptability level of exoskeletons for rehabilitation of the upper limbs from therapists’ perspectives. We analyzed the data collected from a pool of 55 physiotherapists and physiatrists through the distribution of a questionnaire. Pearson’s correlation and multiple linear regression were used for the analysis. The relations between the variables of interest were also investigated depending on participants’ age and experience with technology. The model built from these data demonstrated that the perceived usefulness of a robotic system, in terms of time and effort savings, was the first factor influencing therapists’ willingness to use it. Physiotherapists’ perception of the importance of interacting with an exoskeleton when carrying out an enhanced therapy session increased if survey participants already had experience with this type of rehabilitation technology, while their distrust and the consideration of others’ opinions decreased. The conclusions drawn from our analyses show that we need to invest in making this technology better known to the public—in terms of education and training—if we aim to make exoskeletons genuinely accepted and usable by therapists. In addition, integrating exoskeletons with multi-sensor feedback systems would help provide comprehensive information about the patients’ condition and progress. This can help overcome the gap that a robot creates between a therapist and the patient’s human body, reducing the fear that specialists have of this technology, and this can demonstrate exoskeletons’ utility, thus increasing their perceived level of usefulness.

## 1. Introduction

Upper-limb exoskeletons offer an innovative solution to support the rehabilitation pathway of patients in need of re-educational motor training. They are external structural mechanisms provided with joints and links that are intended to be coupled with those of the human body [1]. Such structures, which are provided with systems of actuators and sensors, are meant to substitute, support, and enhance the activities and movements of the arm when it has been impaired by paralytic effects related to pathologies such as spinal cord injury or stroke. Some examples of exoskeletons for upper-limb rehabilitation are shown in Figure 1.

Their application for rehabilitation purposes is at least comparable, in terms of efficacy, with conventional therapy, and it produces more functional benefits than other kinds of interventions [5]. The key elements for effective rehabilitation therapy include (i) a large amount of practice, (ii) goal-oriented training, (iii) feedback to the patients, (iv) rewarding and interactive exercises, and (v) individualized therapy [6]. The use of exoskeletons guarantees the fulfillment of all of these requirements, allowing the intensive training sessions with specific therapeutic purposes to be carried out while always adapting to the residual motor skills of the patients [7]. Nowadays, despite all of the advantages that we described, exoskeletons are poorly diffused in daily clinical practice [8]. Therapists tend to find them challenging to use and often do not think that robots can offer an actual improvement to the classical therapy that they perform every day. Moreover, they tend to perceive the presence of an exoskeleton as a barrier to their direct contact with the human limb, reducing the feedback on the patient’s conditions. The technology acceptance model (TAM) is a theory that studies the various possible factors influencing users’ acceptance of a certain technology [9]. Introduced by Davis in 1989 [10], the TAM was then expanded and applied in various fields to understand what affects human behavior toward a specific technology, and the acquired knowledge was applied to possibly modify the levels of users’ acceptance or rejection. Other authors have applied the TAM to study users’ intentions to use robotic systems for rehabilitation and assistance, but they always focused only on patients’ points of view [11,12,13]. Therapists, however, are the counterparts of patients, and their opinions on this type of technology can strongly influence its diffusion and use. To the best of our knowledge, no previous studies have been carried out on the acceptability of upper-limb rehabilitation exoskeleton(s) or, in particular, considering therapists as target users. This paper, instead, applies the principles of the TAM to investigate the causes that, according to the therapists’ perspectives, limit the acceptability and, consequently, the use of upper-limb exoskeletons in everyday clinical practice. Data to be fed to the model were collected from a questionnaire that we proposed to a pool of therapists, physiotherapists, and physiatrists. We believe that the investigation of this novel point of view can help identify new methods for improving the quality and usability of robotic systems for rehabilitation.

The rest of this paper is organized as follows. Section 2 describes the state of the art of TAM studies, especially those applied to healthcare technologies. The data collection and analysis process that we used for the construction of our TAM is presented in Section 3. Section 4 presents the results of the work, which are discussed in Section 5. Finally, Section 6 draws the conclusions of the work.

## 2. Related Works

### 2.1. Technology Acceptance Studies

When Davis proposed the TAM, he wanted to understand why people would choose to use a particular technology (such as emails and web processing systems) in the context of their work or daily life. The TAM’s basis comes from physiological theories. The core model by Davis considered two main factors influencing the users’ intentions: *perceived usefulness* (PU) and *perceived ease of use* (EOU) [10,14]. The aim was not to determine whether a technology is actually useful or easy to use, but to understand how potential customers perceive it. This perception is, of course, subject to variations due to age, gender, and experience, which are considered the control variables of the model. The TAM owes its success to the fact that it is an easily understandable and simple model. It is, in any case, subject to wide variations in the correlations among the analyzed variables depending on the users and the system under investigation. Furthermore, it starts from the assumption that human beings are rational in their decisions and behavior, which is not always true [15].

Since its introduction, the TAM has undergone several adaptations, such as extensions to include some “custom variables” in the model. These can be added by each author to better explain the main elements of their TAM [9]. The extensions to the model can be grouped into:External predictors or prior factors: These have a direct effect on the *perceived usefulness* and the *perceived ease of use* variables. They include self-confidence in technology, prior usage, and anxiety towards a technology.Factors coming from other theories: These should increase the reliability of the model. Subjective norms, risk, trust, expectations, and user participation belong to this category.Contextual factors: Gender, technological characteristics, and cultural diversity can influence the global effects of the model.Usage measures: These are related to attitudes toward technology and actual or expected usage of technology according to user’s opinions [9,15].

### 2.2. The TAM Applied to Healthcare Technologies

Even though the TAM was developed for other contexts, it has become progressively more diffused in the healthcare technology field [12]. According to [16], at least 142 empirical studies were conducted on technology acceptance in healthcare by 2021. They mainly dealt with telemedicine, mobile applications, health websites, e-learning in medical education, and electronic health records, and they interviewed nurses, therapists, and patients—especially older people. Some of the most influential factors that they found in those studies were anxiety, computer self-efficacy, innovativeness, and trust. Studies about robotics for healthcare have included a variety of options: social robots, assistive robots, socially assistive robots, telerobots, and telepresence robots [17,18]. Table 1 summarizes works in the literature about the TAM for healthcare robotics. Especially for what concerns the use of rehabilitative and assistive exoskeletons, no study seems to have investigated therapists’ perspectives.

Jankowski and colleagues [11] evaluated long-term changes in technology acceptance during patients’ use of a robotic system for stroke rehabilitation and showed how experience could increase the intention to use the technology. Shore and colleagues [13] proposed a selection of possible TAMs to assess the acceptability level among the elderly with respect to the adoption of assistive exoskeletons in their daily lives. Onofrio and colleagues [12] specifically studied patients’ opinions on the use of upper-limb exoskeletons for assistance in activities of daily living (ADLs). In particular, this study divided the variables influencing the model output into those related to emotional or functional perspectives and into individual or relational ones. PU and EOU, in this sense, were considered individual and connected to the functional perspective. The subjective norm was a relational variable that was connected to both emotional (if coming from relatives and beloved ones) and functional (if coming from clinicians) perspectives. Anxiety, aesthetics, and trust are factors that come from other studies related to individual emotional perspectives. They concluded that for an exoskeleton to be appreciated by patients, the most crucial aspect is that it must be perceived as useful and inspire confidence in the users.

## 3. Methods

### 3.1. A Novel Point of View

Despite the existence of multiple studies dedicated to the acceptability of robotic systems (including those introduced in Section 2.2), we could not find any from the literature that considered physiotherapists as the users to be interviewed in relation to this topic. Our study aims to investigate therapists’ and physiatrists’ perspectives, with the awareness that they, too, are the end users who are asked to interface with exoskeletal technology. Their perception is crucial for guaranteeing the integration of rehabilitation robots into classical therapy sessions.

### 3.2. Data Collection

Data were collected through the distribution of an anonymous questionnaire (see Section A.1). It was distributed both online and in paper form to therapists working in different hospitals in Italy. At the beginning of the survey, we asked the participants to confirm that they belonged to one of the following professional groups: occupational therapists, physiotherapists, or physiatrists. No other eligibility criteria were considered. The data that were collected were anonymized, and the survey was developed according to the law of data protection, according to Art. 13 of the UE 2016/679 norm (General Data Protection Regulation). Its distribution was approved by the Ethical Committee of our university (approval no. 8/2022—16 February 2022).

The questionnaire was composed of twenty-five questions related to the topic of the study. The questions belonged to eight different categories, representing the variables of interest of our TAM:*Intention to use*—ITO: Independent variable and output of the model;*Perceived usefulness*—PU: How useful the therapists perceive an exoskeleton to be in supporting part of their rehabilitation sessions;*Perceived ease of use*—EOU: Level of ease of use of the robotic system in terms of both setting and application during the therapy;*Subjective norm*: The extent to which the opinions and suggestions received from other people (e.g., patients, doctors, and other people who the compiler of the application deems reliable) are favorable to the use of exoskeletons;*Willingness to interact*: How much do the therapists that are interviewed consider it desirable to interact with the system and to be personally involved in the robotic therapy?*Anxiety*: How much do the participants fear that the use of exoskeletons is a source of risk for patients or has negative effects on the therapy?*Time saving*: level of perception of an exoskeleton as helpful in saving time and working with more patients;*Effort saving*: Level of perception of an exoskeleton as helpful in reducing the physical burden on therapists during the execution of rehabilitation exercises.

According to what was introduced in Section 2, the variables representing the core of the TAM are the EOU, PU, and the output, ITO. The other variables that we included belong to the “prior factors” group (*time saving* and *effort saving*) and to the “factors from other studies” group (*anxiety*, *subjective norm*, and *willingness to interact*). Figure 2 shows the structure of the model and the relations among the variables that we proposed.

The following table (Table 2) reports the number of questions referred to each category in the trade-off between the need for a proper number of questions in view of data analysis (i.e., the more, the better) and the total time required to complete the questionnaire (i.e., the less, the better).

The order of presentation of the questions was random and was not related to the categories in order to avoid any possible bias.

Answers were expressed as five levels of agreement with the information provided in each question. They were then converted into numerical values. Scores going from one to five corresponded to the scale of answers from “strongly disagree” to “strongly agree”.

At the beginning of the survey, some additional questions were also proposed with the aim of gathering some personal (age, sex, occupation) and attitude (relationship with the technology, previous experiences with rehabilitation exoskeletons) information from the participants.

### 3.3. Data Analysis

Once we collected all of the answers and we built their correspondence to the numerical scores, we grouped them according to the categories. At this point, we analyzed data as follows according to the process proposed by the literature [23,24,25].

**Cronbach’s alpha**: We evaluated Cronbach’s alpha for each variable of the model. Cronbach’s alpha is a measure of reliability used to assess the internal consistency of the answers given to questions belonging to the same category. Acceptable reliability is represented by values of alpha ranging from around 0.7 to 0.95 [26].**Consistency adjustments**: If some categories obtained alphas lower than 0.7, we further investigated them. We removed the questions that, from an inner correlation study, were revealed to be uncorrelated with the other questions belonging to the same group (under the acceptability threshold of ρ=0.3 for Pearson’s coefficient [27]). If the correlation values were acceptable, we kept the questions in the dataset. We concluded by checking whether defections actually improved the alphas of the various categories.**Pearson’s pairwise correlation**: For every category, we evaluated the mean score from the answers provided by each participant. The literature is unclear about the use of the mean value rather than the median when studying Likert-type categories data [28]. Given that no agreement seems to have been reached, we tried to be consistent with Davis’s work, which carried out TAM analyses by using mean scores, Pearson’s correlation, and multiple linear regression [14,23]. Using the mean data, we built a correlation matrix to highlight the correlations between the variables involved in the TAM. We studied Pearson’s coefficients and their statistical significance through an evaluation of their *p*-values [23].**Multiple regression model**: We created a multiple regression model with the variables of the TAM. ITO was our output variable, while the other categories were the regressors [23].**Effects of control variables**: Given that the control variables could influence the results of our model, we decided to divide the data coming from people who had a previous experience with rehabilitation exoskeletons (for both upper and lower limbs) from the data coming from those who did not. In both cases, we analyzed the correlations between the variables and studied the differences.

## 4. Results

### 4.1. Participants and Answers

Fifty-five people completed the questionnaire. Table 3 shows a summary of the answers that we collected for some questions that we made to characterize the population.

Figure 3 reports a summary of the statistics of the scores attributed to the twenty-five questions of the survey, divided into the eight aforementioned classes.

### 4.2. Results of the Data Analysis

#### 4.2.1. Cronbach’s Alpha and Consistency Adjustments

The analysis of Cronbach’s alpha gave the following results (see Table 4).

The alpha values were acceptable for the ITO, PU, *anxiety*, and *subjective norm*, and they were slightly under the threshold for *time saving*. For all of the categories whose alpha was considered unacceptable, we tested the inner correlations of the answers that we collected. The correlations were evaluated through Pearson’s coefficient. From the evaluation of the acceptability of such correlations, we had to eliminate one of the four questions (and its results) related to the variable *willingness to interact*. As a consequence, Cronbach’s alpha passed from α=0.424 to α=0.759. The alpha could be under the threshold when a category had too few items (i.e., questions). We presented just two items for *effort saving* and *ease of use*, and this could be the cause of their low alpha values. After the correlation analysis, we decided to keep both of the questions for *effort saving* (ρ=0.303>0.3) and to remove the question of the two whose answers had a greater variance for *ease of use* (ρ=−0.0586<0.3).

#### 4.2.2. Pearson’s Correlation

Table 5 reports the results of the analysis of correlation. We have reported in green the correlation coefficients of the couples of variables whose relations are relevant to our TAM.

#### 4.2.3. Effects of Control Variables

As anticipated in Section 2.1, categorical variables can have a strong influence on the results of the analysis.

#### 4.2.4. Experience

We decided to consider the *experience* variable, and we split the dataset into two groups. We divided the results coming from participants who had already used exoskeletons from the data coming from those who had not (see Table 6). As indicated in Table 3, 31 out of 55 therapists and physiatrists declared that they had previously used exoskeletons for their therapy sessions.

The scheme of the TAM with the results of the correlation analysis coming from the two groups is presented in Figure 4.

As can be observed, apart from values related to the correlation between PU and ITO, all of the others significantly changed when isolating data coming from already experienced therapists from data coming from those who had never used an exoskeleton before the questionnaire.

#### 4.2.5. Age

Given the relatively wide range of participants’ ages, we decided to study how the correlations between our variables of interest changed when passing from younger therapists to older ones. From their answers to the general questions, younger therapists seemed more used to the technology (the statistics of the scores that they attributed to questions related to attitude towards technology, presented in Figure 5, confirmed this). They were also those who were more likely to have come into contact with exoskeletons for rehabilitation during their study path. The global age range was 36 years (from 23 to 59 years old). We divided this range into three equivalent sub-ranges (23 ÷ 34 years old, 35 ÷ 46 years old, and 47 ÷ 59 years old) and built a correlation model for the participants belonging to each age group.

When comparing the three models, the values showing an appreciable monotone age-related trend were related to the correlations between *perceived usefulness* and *subjective norms* with the variable *intention to use*. Table 7 shows Pearson’s coefficient values for these two relations.

#### 4.2.6. Multiple Regression Model

When trying to infer cause–consequence relationships, the results of a TAM can also be explained with a regression model. Table 8 reports the beta coefficients of the multiple regression that we modeled on the whole dataset, the standard error, and the results of the F-test, which checked whether the model fit significantly better than a degenerated model consisting of only a constant term. The values of the coefficient of determination and adjusted coefficient of determination of the regression were, respectively, $R^{2}=0.649$ and $R^{2}adj=0.613$. This meant that the model explained approximately 65% of the variability of the response variable *intention to use*. The results were statistically significant, given that $p_{value}=3.89\times 10^{-10}$ (which was under the acceptability threshold). Conversely, we obtained high $p_{values}$ for the pairwise relations of *ease of use*, *willingness to interact*, and *anxiety* with the output variable. These values were, in any case, due to the relatively small sample of participants, and in a future expansion of the study, we can expect to see them be reduced below the acceptability threshold (other works that obtained statistically significant regressions included around 110 participants in their TAMs; see [24]).

We built a second regression model to find the beta coefficients linking *time saving* and *effort saving* (i.e., the prior factors) to *perceived usefulness*. We found that the path coefficients indicating the influences of *time saving* and *effort saving* on PU were, respectively, equal to $\beta_{TS}=0.001606$ and $\beta_{ES}=0.32972$. This second model was also statistically significant ($p_{value}=0.02$; under the threshold), but it suffered from the limited dataset.

## 5. Discussion

The correlation analysis provided information on the percentages of the variance of the latent variables that were explained by the other variables in the model. The correlations that were relevant to our model were all found to be significant. The regression model that we constructed, on the other hand, was statistically reliable overall, but the causal effects of variables such as *willingness to interact*, *anxiety*, and *ease of use* need to be further investigated with additional data to increase the consistency of the results. We hypothesize the following interpretation of the obtained results:As we can see from the global correlation model, the *perceived usefulness* of the exoskeleton explained the majority of the variance of the output (around 77%). This correlation value did not change much when splitting the dataset and comparing the results for the two *experience* subgroups. This point is in agreement with the results obtained from the analysis of patients’ opinions about exoskeletons [12]. The regression coefficient for *perceived usefulness* was β=0.7090, and it had two-tailed significance. We conclude that, from therapists’ perspectives, the first requirement to be considered for application during everyday rehabilitation sessions is to perceive exoskeletons as useful instruments. Nowadays, various benchmarking frameworks are used to evaluate the efficacy of rehabilitation for the motor abilities of neurological patients [29]. They include the use of multiple sensors: from EMG sensors for muscular activation to optoelectronic systems and inertial measurement units for kinematic performance. Applying these systems to the measurement of the improvement of patients who used a rehabilitation robot for their treatment could increase the level of usefulness perceived by therapists.The correlations of *time saving* and *effort saving* with PU showed Pearson’s coefficients that are, respectively, equal to ρ=0.408 and ρ=0.582, and both correlations were significant. The beta coefficient of *effort saving* was statistically significant, while that related to *time saving* was lower than the acceptability threshold of 0.05. Overall, we can infer that therapists tend to find a robotic system that is able to reduce their physical effort in the execution of the rehabilitation exercises slightly more useful than one that makes them save time (i.e., allowing them to treat a patient while a second one uses the exoskeleton for his/her therapy session).The effect of *ease of use* on the output variable in our model proved to be lower than that of *perceived usefulness*. The correlation between *ease of use* and ITO was, in any case, higher for inexperienced therapists, who may have been held back by the prospect of a system that was too complex to learn to use (especially if they did not know about its advantages).The correlation between anxiety towards the technological system and ITO, as can be guessed, was negative (and so was the β coefficient). It is interesting to notice that for participants who had already experienced the use of an exoskeleton for their sessions, the negative effect of the *anxiety* variable on ITO was reduced by about 12%. This information suggests that the use of robotic systems for rehabilitation could be encouraged if therapists have the chance of getting in contact with this kind of technology. Raising the public’s level of knowledge, at least in hospitals and rehabilitation centers, could be a good way to increase the level of confidence in this technology and reduce apprehension in those who do not know how it works. In general, it is important to find methods for reducing the negative impact that the fear of not being able to control the therapy has on the willingness to use robotic systems. As we can understand from the answers collected for **Q7** and **Q9** (see Section A.1), therapists’ *anxiety* was caused by the fact that they felt that they would have no information about how a session conducted by a robot was proceeding if they did not continuously observe the patient. This leads to us losing the advantage in terms of time represented by making one patient use the robot while we work on another patient. An efficient solution to this problem could be investing in complete systems of sensors to be coupled with the exoskeletons and provide reliable and remote feedback to therapists. Other studies proved that feedback is crucial for therapists; rehabilitation experts think that having information about muscular activation and joint positions could be very useful in assessing a patient’s conditions [30]. In this sense, surface electromyography sensors can be integrated into the structure of the robot to record the amount of muscular participation of the patients [31]. Precise position sensors can provide real-time information on the 3D configuration of the arm of the patient. Compact force sensors at the interface with the robot [32] can be used to tune the level of assistance provided by the exoskeleton and assure the therapist that the patient is not harmed. The work described in [33] already moves in this direction; it presented a telerehabilitation system that collected haptic data from the interaction between a patient and a robot and provided them to therapists, who felt confident about being distant from the user while they performed rehabilitation with the device.When studying the results of the relation between the *subjective norm* and the output variable, we could observe that participants’ intention to use exoskeletons had a significant positive correlation with the opinions and suggestions received by doctors and patients (as indicated in the questions that we proposed). The effect of others’ opinions on the use of robots for rehabilitation was reduced by almost 18% for the respondents to the survey who had already used such systems. It also seemed to be reduced when studying the answers of younger therapists, who were more experienced with technologies such as exoskeletons.The questions that we proposed that were related to the *willingness to interact* category aimed to understand if the therapists preferred dealing with robotic systems that gave them many chances for interaction and personalization of the therapy or leaving the exoskeletons in charge of the organization of the entire therapy. We wanted to understand whether it is better to invest in autonomous devices or if it is preferable to find new ways to make therapists cooperate and exchange information with robots. The correlation analysis between *willingness to interact* and *intention to use* produced a Pearson’s coefficient that was statistically irrelevant for inexperienced participants (ρ=0.089<0.1). In any case, the correlation increased by 25.8% when studying the answers provided by therapists who had already used an exoskeleton before compiling the questionnaire (ρ=0.347). We can infer that if inexperienced clinicians prefer the advantages offered by a higher level of automatization of the therapy, therapists who have already come into contact with exoskeletal technology consider collaborating with the system more relevant.

## 6. Conclusions

The study that we conducted aimed, for the first time, to understand the factors that influence the acceptability level of exoskeletons for rehabilitation of upper limbs from therapist’s perspectives. Other works from the literature showed that understanding which factors influence users’ trust and approval towards a certain technology is crucial for improving the quality of human–robot interaction [16,34]. Such studies focused only on patients’ perspectives. With our work, we investigated a new point of view that we believe adds fundamental information for increasing the acceptability and use of rehabilitation robots in clinical environments.

From the analysis of the collected data, we concluded that the perceived level of usefulness was the most relevant aspect influencing users’ willingness to use the technology. The usefulness perception and the level of satisfaction towards the functionalities of rehabilitation technology were demonstrated to increase patients’ trust in robots [35]. Our work confirms that these aspects are also relevant according to therapists and physiatrists. According to our model, the fact that an exoskeleton can reduce the physical effort required of therapists is an element in favor of their perceived utility. In a potential future version of the model, we could look for other possible factors that increase this perception. Both the anxiety produced by the technology and the importance that is given to what other people (even if they are relevant ones) think decrease when analyzing data from people with previous experience with exoskeletons. This is why we see the need to invest in the diffusion of technology and train rehabilitation professionals on the potential that exoskeletons offer. This conclusion is also supported when comparing answers collected by younger therapists with those from older ones. New generations of physiotherapists who have more experience with exoskeletons and often come into contact with them during their studies seem to be less influenced by others’ opinions about this new technology. Our model also supports the conclusion that integrating multi-sensor systems into rehabilitation robots can have an impact on reductions in the effects of *anxiety*, thus increasing therapists’ trust in this technology and augmenting the level of *perceived usefulness*. Coupling joint positions with data coming from electromyography, electroencephalography, and force sensors at the interface between the arm and the exoskeleton can tell a therapist about the user’s level of participation and performance and allow the therapist to monitor their safety. We should invest in new methods for integrating information coming from all of these sensors and make it easily interpretable by therapists. Evaluating it at the end of the therapeutic path can prove the usefulness of the system, while monitoring it in real-time would reassure the therapist about the progress of the robotic sessions while they are busy with other patients, thus possibly increasing the perceived relevance of the *time saving* variable. This study can be improved by introducing new questions into the survey, which can be formulated as clearly as possible to increase the level of inner consistency of the data that we collect. It can also be expanded by finding new therapists and physiatrists to participate in the study. Increasing the number of answers that we gather would also increase the statistical reliability of the model.

## Figures and Tables

**Figure 1 sensors-23-01721-f001:**
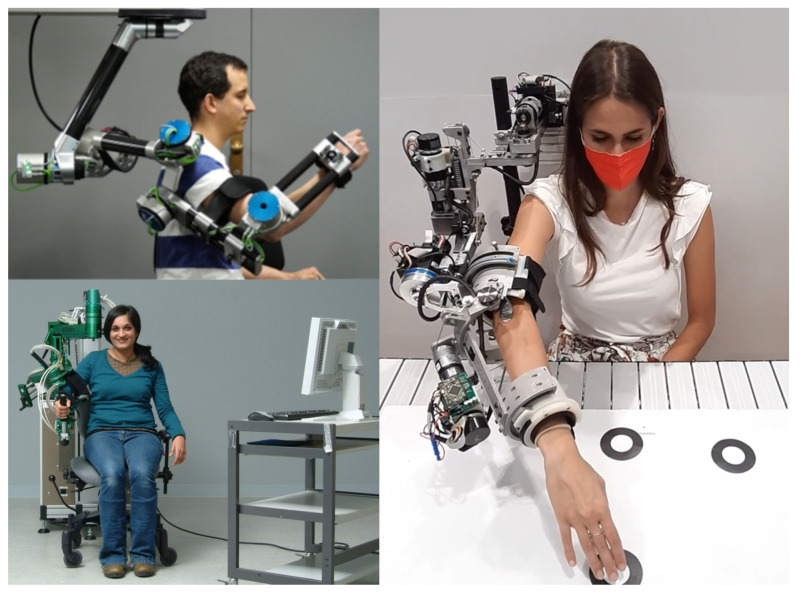
Some examples of upper-limb rehabilitation exoskeletons. The top-left one is ANYexo by ETH Zurich (©2019, Zimmerman et al. from Ref. [2]), the bottom-left one is ARMin (©2010, Nef et al., from Ref. [3]), and the one on the right is AGREE, the prototype from our research group at Politecnico di Milano [4].

**Figure 2 sensors-23-01721-f002:**
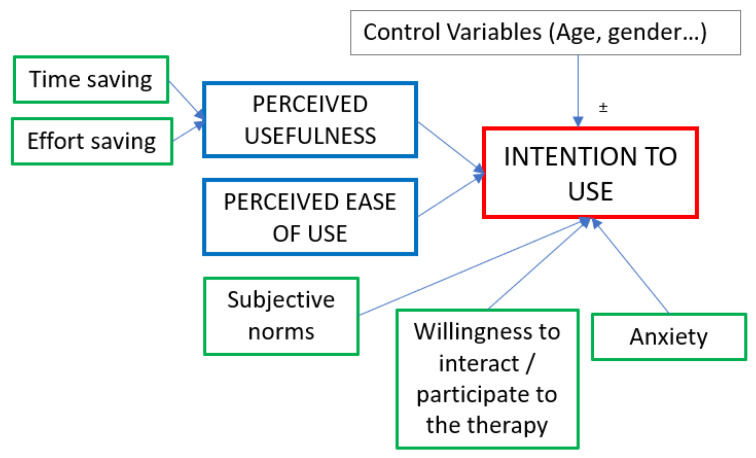
Structure of the TAM. Light blue variables are those of the core of the model, and gray variables are those that we added for our specific study. The dark-blue box represents the output (i.e., the predicted variable).

**Figure 3 sensors-23-01721-f003:**
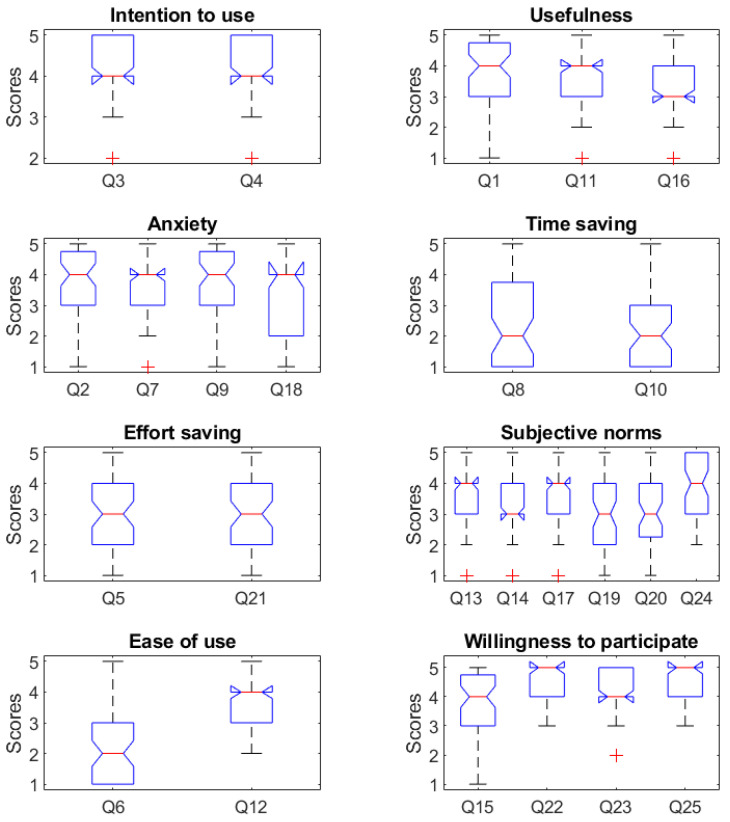
Statistical analysis of the scores given by the 55 users to the questions. Question numbers correspond to those indicated in Section A.1. We gathered the questions by category. In each box plot, the central mark indicates the median, and the bottom and top edges of the box indicate the 25th and 75th percentiles, respectively.

**Figure 4 sensors-23-01721-f004:**
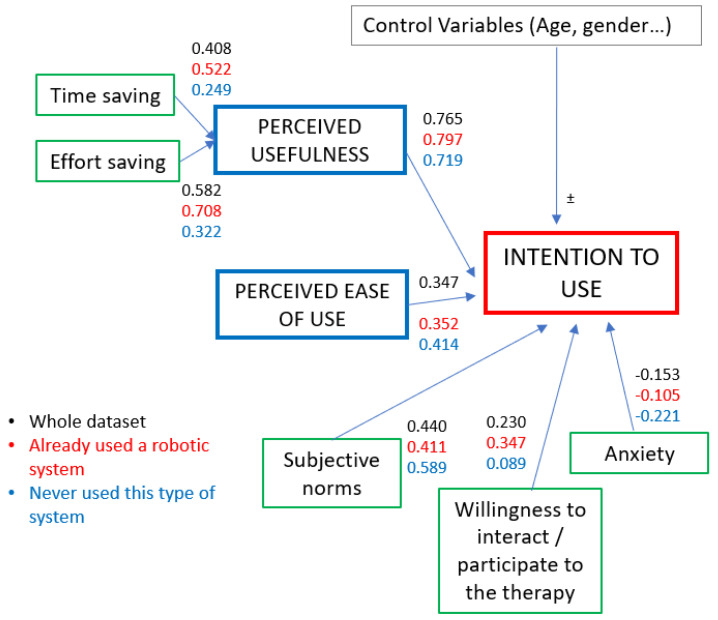
Structure of the TAM with references of the correlations between the various variables involved in the study.

**Figure 5 sensors-23-01721-f005:**
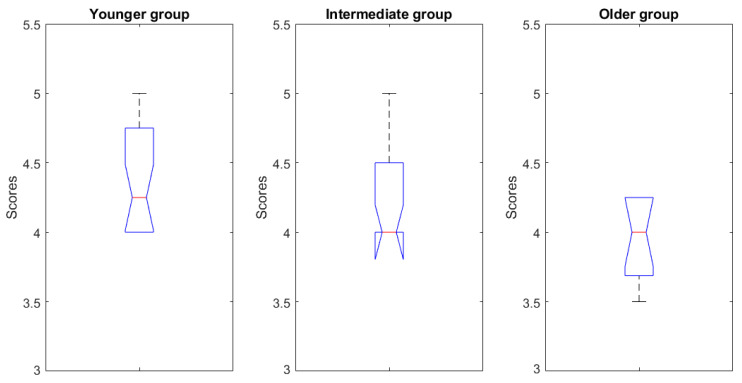
Statistics of the scores given to questions related to participants’ attitudes towards technology, divided according to the three age ranges that we identified.

**Table 1 sensors-23-01721-t001:** Summary of relevant works from the literature investigating applications of the TAM for robotics in healthcare. Types of technologies and interviewed users are indicated in the last two columns.

Study	Author and Date	Technology	Point of View
[19]	He et al., 2022	Social robots in elderly care facilities	Elderly people
[20]	Turja et al., 2020	Service robots	Healthcare professionals (nurses and doctors)
[21]	Nertinger et al., 2022	Remote assistive robots (Humanoids)	Adult patients
[11]	Jankowski et al., 2020	Rehabilitation end-effector (Bi-Manu-Interact robot)	Stroke patients
[22]	Hall et al., 2019	Assistive robots for activities of daily living	Patients
[13]	Shore et al., 2018	Assistive exoskeletons	Elderly people
[12]	Onofrio et al., 2020	Assistive technologies for neurological motor impairments	Neurological patients

**Table 2 sensors-23-01721-t002:** Number of questions in each category of the TAM.

Category (Variable)	No. of Questions
Intention to use	2
Perceived usefulness	3
Anxiety	4
Time saving	2
Effort saving	2
Subjective norm	6
Perceived ease of use	2
Willingness to interact with the system	4

**Table 3 sensors-23-01721-t003:** Summary of the answers to the general questions.

Information	Answer
Age	Mean: 37.4 ± 10.1 y.o. Range: 23–59 y.o. Median: 35 y.o
Gender	• 23 men • 31 women • 1 other
Occupation	• 3 occupational therapists • 3 physiatrists • 1 clinical researcher in physiotheraphy • 48 physiotherapists
Already knew what an exoskeleton is?	• 54 yes • 1 no
Already used an exoskeleton?	• 31 yes • 24 no

**Table 4 sensors-23-01721-t004:** Cronbach’s alpha for the categories of the TAM. ANX: anxiety, ES: effort saving, TS: time saving, SUBJN: subjective norm, WTI: willingness to interact.

	ITO	PU	ANX	TS	ES	SUBN	EOU	WTI
α	0.865	0.829	0.795	0.674	0.464	0.705	−0.123	0.424

**Table 5 sensors-23-01721-t005:** Results of Pearson’s correlation analysis on the dataset. If ** is indicated, the correlation is significant at the 0.01 level. If * is indicated, the correlation is significant at the 0.1 level.

	Int. to Use	Perc. Useful.	Anxiety	Time Saving	Effort Saving	Subj. Norm	Ease of Use	Will. to Interact
Intention to use	1	0.765 **	−0.153 *	0.221	0.359	0.440 **	0.347 **	0.230 *
Perceived Usefulness	0.765	1	−0.273	0.408 **	0.582 **	0.312	0.367	0.168

**Table 6 sensors-23-01721-t006:** Comparison of the correlations between the variables of the model. Global values refer to the analysis of the whole dataset. “Already used” refers to data coming from participants who had already used exoskeletons, and “never used” refers to data coming from those who did.

Independent Variable	Dependent Variable	Pearsons’s Coefficient		
		**Global**	**Already Used**	**Never Used**
Perc. usefulness	ITO	0.765	0.796	0.719
Anxiety	ITO	−0.153	−0.105	−0.221
Subj. norm	ITO	0.440	0.411	0.589
Ease of use	ITO	0.347	0.352	0.414
Will. to interact	ITO	0.230	0.347	0.089
Time saving	PU	0.408	0.522	0.249
Effort saving	PU	0582	0.708	0.322

**Table 7 sensors-23-01721-t007:** Pearson’s coefficients for the correlations between *perceived usefulness* and ITO and between *subjective norms* and ITO for the three age-range groups.

	Younger Group	Intermediate Group	Older Group
Perceived usefulness	0.881	0.722	0.545
Subjective norms	0.290	0.370	0.551

**Table 8 sensors-23-01721-t008:** Results of the construction of a multiple regression model for our data. Significant values are written in bold.

Independent Variable	Dependent Variable	β	SE	tStat	*p* Values
**Perc. Usefulness**	**ITO**	**0.7090**	0.0981	7.2302	$2.9203\times 10^{-9}$
Ease of Use	ITO	0.0228	0.0891	0.2556	0.7993
**Subj. Norm**	**ITO**	**0.2794**	0.1249	2.2423	0.0295
**Will. to Interact**	**ITO**	**0.2098**	0.1341	1.5643	0.0422
Anxiety	ITO	−0.0154	0.0893	−0.1720	0.8642

## Data Availability

The mean answer scores given to all of the questions—as summaries of the data that we collected—are reported in Figure 3. The population-characterizing information is reported in Table 3. If needed, the raw data analyzed in this study are available on request from the corresponding author. The raw data have not been published online to be consistent with the privacy statement that was declared at the beginning of the questionnaire and was approved by the Ethical Committee of Politecnico di Milano.

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
