# Peer review of "Technology Acceptance Model for Exoskeletons for Rehabilitation of the Upper Limbs from Therapists’ Perspectives"

_sensors, 2023, doi:10.3390/s23031721_

Round 1

Reviewer 1 Report

Reviewer 1:

Dear Editor in Chief

This paper investigates a new point of view, different from that of patients. Through the realization of a Technology Acceptance Model, we investigate which factors influence the acceptability level of exoskeletons for rehabilitation of the upper limb from the therapists’ perspective. After reviewing this paper, main comments are listed as:

1-      Novelty and contribution of this paper should be in detail and more highlighted.

2-      In this paper, it should be better to add a table for comparison error such as mean error- mean square error- variance and …. for different control technique.

3-      author suggest to author to add the following new paper in introduction sections. 

1-      New Super-twisting Sliding Mode Control of an Upper Limb Rehabilitation Robot Based on the TLBO Algorithm, Journal of Applied Dynamic Systems and Control 5 (2), 44-53.

4.       Discussions and Conclusions section should be summarized and now it is a large section.

Author Response

We thank the reviewer for reading our paper and giving us suggestions. We will try to address them one by one. Blue portions of the text are the additions/changes we made to the paper.

1. Novelty and contribution of this paper should be in detail and more highlighted.

We appreciate the suggestion of the reviewer. We believe our paper offers a new point of view in the investigation of reasons that limit exoskeletons’ acceptability in daily clinical practice We think this can help improve technology and make it more appreciated and used by therapists. We have rephrased parts of the Abstract and the Introduction section and added some conclusive remarks to the Introduction.

Abstract: The work presented in this paper investigates a novel point of view, different from that of patients, which is normally considered for similar analyses.

Introduction: Other authors have applied TAM to study users' intention to use robotic systems for rehabilitation and assistance, but always focusing just on patients' points of view [11],[12],[13]. […] To the best of our knowledge, no previous study has been carried out about the acceptability of upper-limb rehabilitation exoskeleton(s), in particular considering therapists as target users.

This paper, instead, applies the principles of TAM to investigate the causes that, according to the therapists' perspective, limit the acceptability and consequently the use of upper-limb exoskeletons in everyday clinical practice. […].

We believe the investigation of this novel point of view can help identify new methods to improve the quality and usability of robotic systems for rehabilitation.

We hope this addition helps highlight what we believe is our major contribution.

2. In this paper, it should be better to add a table for comparison error such as mean error- mean square error- variance and …. for different control technique.

We thank the reviewer for the suggestion and we would agree on this point if the focus of the paper was the design of new control techniques.

However, our paper is not focused on new control techniques. We are concerned that the addition of too many details on this would distract the reader from the main goal of the paper, which is the study of factors influencing therapist's acceptability and willingness to use a rehabilitation robot.

Hence, we have decided not to add the above comparison to our paper but, if the reviewer thinks this is an important point, we are open to further discussion

3.Paper suggestion

We are a bit confused by the suggested reference. The paper “New Super-twisting Sliding Mode Control of an‎ Upper Limb Rehabilitation Robot Based on the‎ TLBO Algorithm” describes a novel control mode for a rehabilitation robot based on the TLBO algorithm. Even though it is a very interesting work, our paper does not deal with control modes and we did not consider the effect of different control modalities on technology acceptance according to therapists’ perspectives. Therefore, we have decided not to add this paper to our list, because we believe that the references must be as focused as possible in order for the reader to benefit from a well-organized and synthesized literature review.

We anyway agree that additional references could have been added to the original list of papers. Therefore, we expanded our literature on the existing upper limb exoskeletons.

4. Discussions and Conclusions section should be summarized and now it is a large section

We thank the reviewer for making us notice that the whole section was too long. We decided to divide the final section into two different parts. We slightly changed the organization of the two sections we created in this way. We hope the text is now more readable and it is easier to identify the main points we wanted to highlight in the end part of the paper.

The reviewer suggested extensive English revision. We had the manuscript checked by a native English-speaking colleague and tried to improve our writing style according to his indications. We hope to have achieved some improvement and remain available for any further suggestions.

Reviewer 2 Report

Comment to the authors:

Dear authors:

1- In the title, the authors should mention in some words about the methodology that has applied in this study. Example about “Pearson’s correlation and multiple linear regression”. also, in the keywords.

2- To make the paper more related to the sensors journal, the authors should talk more about the multiple-sensors feedback systems in the abstract, also at least one of the keywords should related to sensors.

3- The authors should include “sensors” in its contributions of this paper to be more related to the sensors journal.

4- In the abstract, Technology Acceptance Model in line 5 should be written “Technology acceptance model”.

5- The number of references should be increased to be at least more than 30 reference.

6- The authors should add a figure of “exoskeletons for rehabilitation of the upper limb”.

7- In line 37, the following sentence: “TAM (Technology Acceptance Model) is a theory studying the various possible factors influencing users’ acceptance of a certain technology[6]. should be written as “Technology Acceptance Model (TAM) is a theory studying the various possible factors influencing users’ acceptance of a certain technology[6].

8- In the end of the introduction section, the authors should write an overview about the organization of the rest of the paper. For example, the rest of the paper is organized as follows. In section 2, the related work is introduced….etc.

9- In the related work section, i.e. in section 2, the authors should add a table to summarize the related work and what this study will added.

10- the title of section 3, in line 103, is too long “3. Study of technology acceptance from a novel point of view”.

11- section 3 is too short. It is not appropriate to write a section of only 1 paragraph. The authors should increase the contents of it.

12- in line 113, the author mentioned “It was distributed both online and in paper form to therapists working in different hospitals in our country.” . which country?

13- in line 141, the word “section” should be used instead of “Par.”

14- regarding the caption of Figure 1. The authors wrote” Figure 1. TAM structure. Light blue variables are those of the core of the model, green variables are the ones we added for our specific study. The dark-blue box represents the output (i.e., predicted variable)”.

Where are the green variables?!!!!!

Author Response

Dear reviewer,

we thank you for taking the time to read our paper and for your kind observations and suggestions. We have tried to address the points you highlighted one by one:

1.In the title, the authors should mention in some words about the methodology that has applied in this study. Example about “Pearson’s correlation and multiple linear regression”. also, in the keywords.

We thank the reviewer for the suggestion. While we agree that words like “Pearson’s correlation and multiple linear regression” could be helpful to put more emphasis on the methods, we prefer to keep a short title and move the focus on the methods in the abstract and introduction. We believe that this improves readability and makes the introduction more complete on the methods.

We hope that the expression “Technology Acceptance Model”, which describes a method that includes the use of linear regression and Pearson’s correlation in its development, can be considered somehow self-explanatory. We added the suggested keywords to the list and we hope that, together with the abstract, they can rapidly clarify possible doubts of the readers. We are anyway open to further suggestions from the reviewer.

2. To make the paper more related to the sensors journal, the authors should talk more about the multiple-sensors feedback systems in the abstract, also at least one of the keywords should related to sensors.

We thank the reviewer for the observation and agree with him/her. We added the keyword “integrated sensor systems” to the list, and modified the abstract as follows: “The conclusions drawn from our analyses show that we need to invest in making this technology better known to the public, in terms of education and training if we aim at making exoskeletons genuinely accepted and usable by therapists. Also, integrating exoskeletons with multiple-sensors feedback systems would help provide comprehensive information about the patient's condition e progresses. This can help overcome the gap that the robot creates between the therapist and the patient's human body, reducing the fear specialists have towards this technology, and can demonstrate exoskeletons' utility, increasing the perceived level of usefulness.” Lines 19-23

We hope these changes in the abstract help focus more on the sensors topic and highlight the importance of reconstructing comprehensive feedback about the conditions of the patient towards many integrated sensors.

3.The authors should include “sensors” in its contributions of this paper to be more related to the sensors journal.

We thank the reviewer for the recommendation.

We are aware that we did not directly use specific sensors to realize this study, but we believe that the results of our model highlight the importance of adding multiple sensor systems to exoskeletons and other similar robotic devices if we want to make them effective and usable. Feedback about the patient's condition is fundamental to increasing the level of trust in therapists and decreasing the gap they perceive between them and the patient’s arm due to the robot's presence. 

Moreover, benchmarking methods that involve the use of EMG and EEG sensors and evaluate the kinematic performances of the patients are currently applied to measure the improvement of the patients after a period of rehabilitation. Seeing that their patients improved by using an exoskeleton for their therapy can surely increase the perceived usefulness from therapists' point of view.

We tried to further address these points by adding some observations to the Introduction (“Moreover, they tend to perceive the presence of an exoskeleton as a barrier to their direct contact with the human limb, reducing the feedback on the patient's conditions.” line 44-46) and to the Discussion sections.

We tried to deepen the description of fundamental sensors that are needed to provide comprehensive feedback to the therapist, adding references. The discussion section now includes the following observations related to the effect of sensors-systems on the variable "Anxiety" (Lines 329-343):  “ An efficient solution to this problem could be investing in complete systems of sensors to be coupled to the exoskeletons and provide reliable and remote feedback to the therapists. Other studies proved that feedback is crucial for therapists: rehabilitation experts think having information about muscular activation and joint positions could be very useful to evaluate the conditions of a patient [ 29]. In this sense, surface electromyography sensors can be integrated into the structure of the robot to record the amount of muscular participation of the patients [30]. Precise
position sensors can provide real-time information about the 3D configuration of the arm of the patient. Compact force sensors at the interface with the robot [31 ] can be used to tune the level of assistance provided by the exoskeleton and assure the therapist that the patient is not harmed. The work presented in [ 32 ] already moves in this direction: it presents a telerehabilitation system that collects haptic data from the interaction between a patient and a robot and provides them to therapists, who feel confident about being distant from the user while he/she performs rehabilitation with the device.

Regarding the effects on the variable "perceived usefulness" we wrote: Nowadays, various benchmarking frameworks are used to evaluate the efficacy of rehabilitation on the motor abilities of neurological patients [29]. They include the use of multiple sensors: from EMG sensors for muscular activation to optoelectronic systems and inertial measurement units for kinematic performances. Applying these systems to measure the improvement of patients who used a rehabilitation robot for their treatment could increase the level of usefulness perceived by therapists. (line 294-300)

The concept is then summarized in the Conclusions section - lines 389-400 (“Our model supports also the conclusion that integrating multiple-sensor systems into the rehabilitation robots can be of impact in reducing the effect of Anxiety, increasing therapists' trust towards this technology and augmenting the level of Perceived usefulness. Coupling joint positions with data coming from electromyography, electroencephalography and force sensors at the interface between the arm and the exoskeleton can tell the therapist about the user's level of participation and performance and monitor his/her safety. We should invest in new methods to integrate information coming from all these sensors and make it easily interpretable by therapists. Evaluating it at the end of the therapy path can prove the usefulness of the system while monitoring it in real-time would reassure the therapist about the progress of the robotic sessions, while he is busy with other patients, possibly increasing the perceived relevance of the \textit{Time Saving} variable.”).

We tried to create a common thread between the various sections and hope this new organization of the way we presented the importance of sensors for exoskeletons application helps highlight the relevance of the topic.

4. In the abstract, Technology Acceptance Model in line 5 should be written “Technology acceptance model”.

We thank the reviewer for making us notice that: we changed the spelling.

5. The number of references should be increased to be at least more than 30 reference.

We thank the reviewer to make us notice we had to expand our references and strongly agree with his/her suggestion. We added some references about rehabilitation exoskeletons existing in the literature (references 2, 3, 4), about TAM studies related to robotics in healthcare (21, 22,  19, 20), about the importance of sensors for benchmarking methods (29) and about types of feedback that therapists consider fundamental (30, 31, 32, 33,). We hope now our references list is more complete.

6. The authors should add a figure of “exoskeletons for rehabilitation of the upper limb”.

We thank the reviewer for the suggestion. We added an image in the Introduction paragraph. It represents three different upper-limb rehabilitation exoskeletons, one of which is the prototype we developed at Politecnico di Milano. 

7. In line 37, the following sentence: “TAM (Technology Acceptance Model) is a theory studying the various possible factors influencing users’ acceptance of a certain technology[6].“  should be written as “Technology Acceptance Model (TAM) is a theory studying the various possible factors influencing users’ acceptance of a certain technology[6].”

We agree with the reviewer and thank him/her for the observation. We are sorry for the oversight. We modified the sentence in the introduction as he/she suggested.

8. In the end of the introduction section, the authors should write an overview about the organization of the rest of the paper. For example, the rest of the paper is organized as follows. In section 2, the related work is introduced….etc.

We thank the reviewer for the suggestion. We introduced the description of the organization of the paper at the end of the introduction section: “The rest of the paper is organized as follows. Section 2 describes the state of the art of TAM studies, especially applied to healthcare technologies. The data collection and analysis process we actuated for the construction of our TAM is presented in Section 3. Section 4 presents the results of the work, which are discussed in Section 5. Finally, Section 6 draws the conclusions of the work.” Lines60-64

9. In the related work section, i.e. in section 2, the authors should add a table to summarize the related work and what this study will added.

We thank the reviewer and agree with his/her suggestion. We integrated a table summarizing some state-of-the-art works that applied TAM for healthcare robotics. We highlighted the type of technology investigated and the target subjects of the analysis and tried to report at least a study for each kind of technology we identified under the definition of robotics for healthcare. We underlined that no other study conducted about exoskeletons for rehabilitation and assistance investigates therapists’ points of view (“Tab. summarizes literature works about TAM for healthcare robotics. Especially for what concerns the use of rehabilitation and assistive exoskeletons, no study seems to have investigated therapists' perspectives.”). lines 101-103

10. the title of section 3, in line 103, is too long “3. Study of technology acceptance from a novel point of view” + 11. section 3 is too short. It is not appropriate to write a section of only 1 paragraph. The authors should increase the contents of it.

We have tried addressing together points 10 and 11 and thank the reviewer for the suggestion. We have decided to include Section 3 in the section “Methods”, as a subsection. We changed the title to make it shorter: “A novel point of view”. We hope these changes made the whole structure more coherent and the text well distributed.

12. in line 113, the author mentioned “It was distributed both online and in paper form to therapists working in different hospitals in our country.” . which country?

Thanks for underlying this point: we changed the text and specified that the country in which we distributed the questionnaire is Italy.

13. in line 141, the word “section” should be used instead of “Par.”

We thank the reviewer for highlighting this point, we agree that Section is the correct term to be used in this sentence.   

14.regarding the caption of Figure 1. The authors wrote” Figure 1. TAM structure. Light blue variables are those of the core of the model, green variables are the ones we added for our specific study. The dark-blue box represents the output (i.e., predicted variable)”.Where are the green variables?!!!!!

Thanks for highlighting this mistake. We corrected the word inserting the real colour we meant to write (“grey”) and are sorry for the oversight.

We will upload the updated document and a pdf produced with a change-tracking system to make them easily identifiable. 

Reviewer 3 Report

The subject of the paper is most interesting. It is well structured, and cleared explained. There is a good number of recent references, as well as those related with the state of the art and to fundamental concepts related to the topic, used in a consistent way. The abstract and the introduction clearly present the problem and there are a detailed conclusion section. The paper is written in good english.

Only one minor suggestion is, in 4.1, when authors write “our country” you should write the country name (Italy) because it gains in clarity.  

Author Response

We thank the reviewer for reading our work and for his/her positive comments. We are glad he/she appreciated our work.

We thank him/her for the observation and proceeded by changing the term “our country” to “Italy”.

We will upload the updated document and a pdf produced with a change tracking system to make them easily identifiable.

Round 2

Reviewer 2 Report

Thank you for answering  round 1 comments.